# Unveiling Adulteration in Herbal Markets: MassARRAY iPLEX Assay for Accurate Identification of *Plumbago indica* L.

**DOI:** 10.3390/ijms26157168

**Published:** 2025-07-24

**Authors:** Kannika Thongkhao, Aekkhaluck Intharuksa, Ampai Phrutivorapongkul

**Affiliations:** 1School of Languages and General Education, Walailak University, Nakhon Si Thammarat 80160, Thailand; kannika.to@wu.ac.th; 2Herbology Research Center, Walailak University, Nakhon Si Thammarat 80160, Thailand; 3Department of Pharmaceutical Sciences, Faculty of Pharmacy, Chiang Mai University, Chiang Mai 50200, Thailand; ampai.phrutiv@cmu.ac.th

**Keywords:** DNA barcode, identification, leadwort, MassARRAY, *Plumbago*, raw material, single nucleotide polymorphism (SNP)

## Abstract

The root of *Plumbago indica* L. is commercially available in herbal markets in both crude and powdered forms. *P. indica* root is a key ingredient in numerous polyherbal formulations. However, *P. indica* has two closely related species, *P. zeylanica* L. and *P. auriculata* Lam. Since only *P. indica* is traditionally used in Thai polyherbal products, adulteration with other species could potentially compromise the therapeutic efficacy and overall effectiveness of these formulations. To address this issue, a MassARRAY iPLEX assay was developed to accurately identify and differentiate *P. indica* from its closely related species. Five single nucleotide polymorphism (SNP) sites—positions 18, 112, 577, 623, and 652—within the internal transcribed spacer (ITS) region were selected as genetic markers for species identification. The assay demonstrated high accuracy in identifying *P. indica* and was capable of detecting the species at DNA concentrations as low as 0.01 ng/µL. Additionally, the assay successfully identified *P. zeylanica* in commercial crude drug samples, highlighting potential instances of adulteration. Furthermore, it was able to distinguish *P. indica* in mixed samples containing *P. indica,* along with either *P. zeylanica* or *P. auriculata*. The developed MassARRAY iPLEX assay proves to be a reliable and effective molecular tool for authenticating *P. indica* raw materials. Its application holds significant potential for ensuring the integrity of herbal products by preventing misidentification and adulteration.

## 1. Introduction

*Plumbago indica* L. (Indian leadwort or Jettamoon Pleung Daeng in Thai) is a medicinal plant belonging to the Plumbaginaceae family (Figure 1A–C). It has been widely utilized in traditional medical systems, including Ayurvedic, Chinese, and Thai traditional medicine. The roots and leaves of *P. indica* have been traditionally used for their therapeutic properties, including digestive stimulation, anti-inflammatory effects, anthelmintic activity, diaphoretic properties, and expectorant benefits [1]. Scientific evidence suggests that the ethanolic extract of *P. indica* exhibits strong genotoxic effects and suppresses the cell cycle in human lymphocytes [2]. Similar to other species in the Plumbaginaceae family, the roots of *P. indica* contain plumbagin (5-hydroxy-2-methyl-1,4-naphthoquinone), which is the primary bioactive compound among several other phytochemicals present in the plant. Plumbagin plays a crucial role in anti-inflammatory, antibacterial, antioxidant, and anticancer activities. Its antioxidant and anti-inflammatory properties have shown promising potential for the treatment of neurodegenerative and cardiovascular diseases [3]. The roots of *P. indica* (Figure 1C,D) are highly valued in the pharmaceutical and natural medicine industries due to the pharmacological properties of plumbagin, which influence the market dynamics, utilization, and pricing of herbal raw materials [1]. *P. indica* is a key component in various polyherbal medicinal formulations, including “Ya Benchakun,” “Ya Fai Pralaikan,” “Ya Fai Ha Kong,” and “Ya Lueat Ngam”—all of which are recorded in the National List of Essential Medicines in Thailand. For instance, “Ya Benchakun” (Figure 1E) is composed of five medicinal plant species: the roots of *P. indica*, the stem of *Piper interuptum* Opiz., the fruits of *Piper longum* L., the roots of *Piper sarmentosum* Roxb., and the rhizomes of *Zingiber officinale* Roscoe [4]. This formulation is available in various herbal dosage forms, including powders, tablets, pills, and infusions. Scientific evidence suggests that “Ya Benchakun” holds therapeutic potential for the treatment of cholangiocarcinoma [5]. Additionally, the ethanolic extract of “Ya Benchakun” has demonstrated anti-allergic and anti-inflammatory effects by inhibiting nitric oxide (NO) production [4]. Another polyherbal formulation, “Ya Lueat Ngam,” is traditionally used to treat primary dysmenorrhea, pain, and inflammation [6]. In addition to *P. indica*, two other *Plumbago* species are found in Thailand: *P. zeylanica* L. (Ceylon leadwort or Jettamoon Pleung Kaow in Thai) and *P. auriculata* Lam. (Cape leadwort or Jettamoon Pleung Farang). *P. zeylanica* has been reported to exhibit significant therapeutic potential for managing diabetes, cardiovascular disorders, ulcers, liver problems, obesity, wound healing, and cancer [7]. Meanwhile, *P. auriculata* Lam. is primarily used as an ornamental plant, though its hydroalcoholic extract has shown promising antiparasitic activity, as well as anti-inflammatory properties, similar to other species within this genus [8,9]. However, among these species, only the roots of *P. indica* are used as an ingredient in the Thai polyherbal formulations mentioned above. The substitution of *P. indica* with alternative species may compromise the therapeutic efficacy of these formulations, potentially affecting their intended medicinal benefits.

The increasing demand for herbal materials in the global herbal market extends beyond *Plumbago indica* to other valuable medicinal plants, such as *Centella asiatica* (L.) Urb., *Cyanthillium cinereum* (L.) H. Rob., *Saraca asoca* (Roxb.) W.J. de Wilde, and *Myristica fragrans* Houtt. [10,11,12,13]. Scientific evidence indicates that 1-year-old *P. indica* roots cultivated under conventional field conditions can yield 1.33 g of plumbagin per 100 g (dry weight) of raw material [14]. This raises concerns regarding product quality, particularly in relation to unauthenticated practices, such as adulteration and product substitution [15]. To mitigate these issues, the World Health Organization (WHO) has established internationally recognized guidelines for assessing raw materials used in herbal medicine, ensuring their quality, efficacy, and safety [16]. Various authentication tools have been developed to prevent adulteration and substitution. Morphological identification relies on characteristics such as leaf shape, flower structure, stem morphology, bark features, and fruit and seed traits to distinguish herbal materials [17]. Microscopic analysis involves the use of microscopy to examine cellular structures and anatomical features unique to specific plant species [18]. Chemical profiling methods analyze the chemical composition of herbal materials through techniques such as high-performance liquid chromatography (HPLC) [19,20], gas chromatography-mass spectrometry (GC-MS) [21,22], liquid chromatography-tandem mass spectrometry (LC-MS/MS) [20,23], and high-performance thin-layer chromatography (HPTLC) [20]. In addition, DNA barcoding has emerged as a highly effective tool for the quality control of herbal materials. This method enables species identification by analyzing single-nucleotide polymorphisms (SNPs) within short DNA regions, making it applicable to both fresh and highly processed herbal materials [24,25]. DNA barcoding has been integrated with various advanced technologies to enhance species differentiation, including next-generation sequencing (NGS), lateral-flow immunochromatographic assays (LFA), and high-performance thin-layer chromatography (HPTLC) [26,27,28]. Recently, the MassARRAY iPLEX method has been employed for SNP identification in plant species. For example, it has been used to analyze embryo DNA to reveal the abscission of self-fertilized progeny during fruit development in macadamia, as well as for species differentiation in plants such as common bean and Moscow salsify [29,30,31]. This method combines matrix-assisted laser desorption/ionization time-of-flight mass spectrometry (MALDI-TOF MS) with PCR amplification, utilizing single-base extension (SBE) chemistry and dideoxynucleotides (ddNTPs) to incorporate a single base at the 3′-end of an extension primer. The mass of the incorporated nucleotide is then measured via mass spectrometry, allowing for the precise identification of nucleotide sequences based on their specific molecular mass [32]. However, the application of MassARRAY technology for the quality assessment of medicinal plants through identification or authentication remains relatively limited.

Our previous DNA barcoding study on *P. indica*, *P. zeylanica*, and *P. auriculata* identified nucleotide variation sites within the internal transcribed spacer 2 (ITS2) region, which has been proposed as an effective DNA barcode for authenticating *Plumbago* crude drugs [33]. However, species identification using the DNA barcoding method is time-consuming, as it requires nucleotide sequencing, Basic Local Alignment Search Tool (BLAST) searches (https://blast.ncbi.nlm.nih.gov/Blast.cgi, accessed on 5 June 2025), and subsequent data analysis. As an alternative, the MassARRAY technique offers a DNA sequencing-free approach with several advantages. This method supports multiplex reactions within a single run and provides higher sensitivity and accuracy, high throughput capability, cost-effectiveness, flexibility, and reliability compared to conventional DNA sequencing methods [34]. Although this technique has been widely applied for SNP identification in various fields, its application in plant species identification is still limited. Therefore, the objective of this study is to apply MassARRAY technology for *Plumbago* species identification using DNA barcode information from our previous study. Specifically, this method will be employed to identify *P. indica* raw materials, detect SNP sites along the ITS barcode, and differentiate *P. indica* from its closely related species, *P. zeylanica* and *P. auriculata*. By demonstrating the feasibility of MassARRAY for herbal material authentication, this study aims to expand its application in the field of herbal quality control.

## 2. Results

### 2.1. MALDI-TOF MS Analysis Differentiated P. indica from P. zeylanica and P. auriculata

The molecular masses obtained from the extension products were analyzed using MALDI-TOF MS analysis. Five unextended primers (UEPs), namely P1#1, P1#2, P2#1, P2#2, and P2#3, exhibited masses of 6414.2, 4599.0, 5172.4, 4625.0, and 5811.8 Daltons (Da), respectively. After the iPLEX extension process, the mass of each extension primer varied due to the contribution of the nucleotide bases (dCMP, dTMP, dAMP, or dGMP) at the 3ʹ-end of each extension primer (Table 1). The MALDI-TOF MS spectra exhibited distinct mass profiles for authentic *P. indica*, *P. zeylanica*, and *P. auriculata* (Figure 2). The mass profile of *P. indica* extension products was obtained from all five extension primers. The extension products revealed molecular masses of 6741.3 Da (P1#1), 4846.2 Da (P1#2), 5459.6 Da (P2#1), 4952.1 Da (P2#2), and 6059.0 Da (P2#3), corresponding to the nucleotide bases T, G, G, indel (insertion/deletion: indel is a biological term referring to an insertion or deletion of nucleotides within a DNA sequence), and C at positions 18, 112, 577, 623, and 659, respectively. The mass spectra of *P. zeylanica* revealed molecular masses of 6741.3 Da (P1#1), 4846.2 Da (P1#2), 5459.6 Da (P2#1), 4912.2 Da (P2#2), and 6059.0 Da (P2#3), corresponding to the bases T, G, G, C, and C, respectively. An additional mass at 4870.2 Da (base T) was observed from the P1#2 extension product. The mass profile of *P. auriculata* exhibited spectra at 6661.4 Da (P1#1), 4870.2 Da (P1#2), 5499.5 Da (P2#1), 4912.2 Da (P2#2), and 6059.0 Da (P2#3), which indicated nucleotide bases C, T, T, C, and C, respectively. No additional peaks were observed from the *P. auriculata* samples (Table 2).

### 2.2. Sensitivity of MassArray Technique for the Identification of P. indica, P. zeylanica, and P. auriculata

Sensitivity analysis of all iPLEX reactions using the extension primers (P1#1, P1#2, P2#1, P2#2, and P2#3) showed that detection sensitivity depended on genomic DNA concentrations (Table 3). The iPLEX extension primers produced accurate nucleotide sequence results when genomic DNA templates were within the range of 10–0.01 ng/µL. Genomic DNA concentrations lower than 0.01 ng/µL resulted in incorrect nucleotide sequence detection for *Plumbago* species.

### 2.3. Identification of P. indica in Plumbago-Mixed Samples the MassARRAY Analysis Successfully Identified Plumbago indica in the Mixed Plumbago Sample

In *Plumbago*-mixed samples, the iPLEX extension reaction successfully identified the mixture of *P. indica* and *P. auriculata* (Figure 3). Mass spectra revealed the indel SNP at position 623 for *P. indica* (Figure 3A), which was detected in all *P. indica* mixed samples (Figure 3A,C,D). Species-specific SNPs for *P. auriculata* at positions 18, 112, and 577 were observed in all *P. auriculata*-mixed samples (Figure 3A,B,D). In the *P. indica*+*P. zeylanica* reaction, the iPLEX extension product from the P1#2 primer showed two spectra at 4846.2 and 4870.2, indicating the presence of bases G and T (Figure 3C). For the three-species mixed sample, the indel SNP of *P. indica* was detected along with all three *P. auriculata* species-specific SNPs. The internal control base at SNP position 652, detected by the P2#3 primer, was present in all samples, and the nucleotide sequence was correctly identified through MassARRAY analysis (Figure 3).

### 2.4. Identification of Jettamoon Pleung Daeng Crude Drugs and the Crude Drug Composed in the Traditional Thai Medicinal Formulations

This study focused on the authentication of commercial Jettamoon Pleung Daeng crude drugs derived from *P. indica* root (C-1 to C-9) and traditional Thai medicinal formulations containing Jettamoon Pleung Daeng crude drug as an ingredient. Both purchased and in-house prepared formulations were analyzed to verify the presence of Jettamoon Pleung Daeng in these preparations. The results, shown in Table 4, confirmed that all Jettamoon Pleung Daeng crude drug samples originated from roots of *P. indica*, as all iPLEX extension products exhibited identical SNP positions compared to the mass profile of the authentic *P. indica* species. However, MassARRAY analysis revealed that the commercial polyherbal formulations “Ya Benchakun” (R-2) and “Ya Fai Ha Kong” (R-6) were adulterated, as the *P. indica* species in these polyherbals was replaced by *P. zeylanica*.

## 3. Discussion

Our MassARRAY iPLEX results identified the presence of the roots *P. zeylanica* in commercial herbal products that contained Jettamoon Pleung Daeng crude drugs, indicating that adulterated products are still prevalent in the market. This highlights the importance of authentication in herbal medicine raw materials to ensure the safety, efficacy, and quality of herbal products. While the WHO and national FDA authorities have taken these issues seriously, as reflected in their policies and regulations, adulterant and substitute products continue to appear both online and on shelves. In 2019, a comprehensive global survey using DNA-based authentication methods revealed that 27% of 5957 commercial herbal products sold across 37 countries were adulterated [35]. Although *P. indica*, *P. zeylanica*, and *P. auriculata* all contain plumbagin, the concentration of this compound, along with other potentially differing compounds, can vary between species and may alter their therapeutic effects [36]. As a result, substituting one species for another without consulting a qualified traditional medicine expert is not recommended.

As the cost of next-generation sequencing continues to decrease, DNA sequencing-based methods are becoming more cost-effective. However, DNA degradation can affect the accuracy of DNA-based analysis. Furthermore, the quality of results obtained from DNA barcoding and metabarcoding methods depends heavily on the quality of the DNA template and the completeness of the reference database used for accurate herbal species identification. Several studies have utilized SNP sites within DNA barcode sequences, combining them with high-sensitivity, cost-effective, and rapid technologies to make DNA barcoding more efficient and accessible [26,37,38]. SNP-based identification provides a precise and reliable method for species identification and quality control, which is especially valuable for distinguishing closely related plant species. In this study, we utilized SNP sites within the ITS barcode region for iPLEX extension assay design, based on findings from our previous work. The mass of the iPLEX extension products was determined using the MassARRAY system for accurate and high-throughput analysis. The MassARRAY iPLEX technique does not require a nucleotide sequence database for analysis, nor does it rely on fluorescent compounds. Instead, it directly analyzes DNA products, making it a versatile method as the assays are not pre-spotted onto the chip by the manufacturer [39]. In terms of sensitivity, MassARRAY iPLEX technique provides high sensitivity, capable of detecting nucleotides at concentrations as low as 0.01 ng/µL, which is 1000-fold sensitive compared to the gold standard sanger Sequencing method, which requires a minimum DNA template concentration of 10 ng/µL using this primer set. This high sensitivity is beneficial for analyzing degraded DNA templates, which are commonly found in the powdered herbal products.

In this study, the application of the MassARRAY iPLEX for *P. indica* demonstrated species-specific spectra across all targeted SNP sites, confirming that the iPLEX extension primers are highly effective for identifying *P. indica* in both single-species and mixed-species samples. However, the mass of the P1#2 extension product, with a peak at 4870.2 Da, appeared in both *P. auriculata* and *P. zeylanica*. Sanger sequencing results showed that *P. zeylanica* had the heterozygous site, G or T, at this position (Appendix A). This suggests that the P1#2 primer, which targets nucleotide position 112, was not suitable for the species-specific site of *P. auriculata*. This finding may be attributed to the specific DNA barcode region used in this study, as heterozygosity and loss of heterozygosity within the ITS region could complicate the MassArray analysis [40,41]. Therefore, it is crucial to consider the potential impacts of heterozygosity and loss of heterozygosity (LOH) on the accuracy and reliability of the results.

Molecular techniques have become indispensable for identifying plant species in crude drugs and herbal products. Their widespread adoption stems from their high accuracy, reliability, and rapid turnaround time, offering a significant advantage over conventional methods like organoleptic, macroscopic, microscopic, and chemical analyses, which often struggle to distinguish closely related plant species [42,43,44,45]. However, a primary limitation of molecular techniques lies in their reliance on high numbers of secondary metabolites and the integrity of genomic DNA [45]. DNA degradation can commonly occur during various stages, including the collection, processing, and storage of herbal materials, as well as during the preparation of traditional formulations, particularly when high-temperature treatments are involved [46]. This degradation can impede successful amplification and sequencing, thereby compromising the effectiveness of molecular assays. To overcome these challenges and ensure accurate species identification, especially when molecular approaches face technical limitations, integrating orthogonal methods such as chemical profiling is recommended to validate molecular findings [44,45,47]. This complementary approach enhances the overall reliability and robustness of species authentication in herbal medicine research and quality control.

## 4. Materials and Methods

### 4.1. Plant Materials, Commercial Crude Drugs, and Traditional Formulations

Three species of *Plumbago*—*P. indica* L., *P. zeylanica* L., and *P. auriculata* Lam.—were randomly collected and used as reference specimens. Details of the collection sites, dates, and code numbers are provided in Table 5. All plant specimens were identified by Miss Wannaree Charoensup, a botanist at the Faculty of Pharmacy, Chiang Mai University, Thailand. The voucher specimens were subsequently deposited in the official herbarium of the Faculty of Pharmacy, Chiang Mai University. For molecular analysis, leaf samples were placed in sterile polyethylene bags containing silica gel to preserve DNA for further extraction. Additionally, commercial crude drugs derived from *P. indica* root, along with Thai traditional formulations containing *P. indica* root—namely “Ya Benchakun,” “Ya Fai Ha Gong,” and “Ya Fai Pralaikan”—were both prepared in-house and obtained from herbal dispensaries. The in-house polyherbal formulations were prepared following the guidelines of the National List of Herbal Medicines, Thailand. All collected samples were subsequently subjected to an authentication test (Table 5).

### 4.2. Primer Sets and Design

Nucleotide sequences of *P. indica*, *P. zeylanica*, and *P. auriculata* from our previous study were used as reference templates for sequence analysis. In this study, five SNP sites (positions 18, 112, 577, 623, and 652) were selected as targeted markers to differentiate among the three *Plumbago* species (Figure 4A). Among these, the SNP site at position 652 was designated as an internal control for the assay. For *P. indica*, G112, G577, and indel623 were identified as potential species-specific markers. Notably, *P. indica* exhibited two species-specific SNPs, with nucleotide G at position 112 and an insertion/deletion (indel) at position 623. For *P. zeylanica*, T18, T112, G577, and C623 were selected as differentiating markers. For *P. auriculata*, C18, T112, T577, and C623 were identified as the distinguishing SNPs, with C18 and T577 serving as its species-specific markers.

Three sets of primers were designed and applied for *P. indica* identification using the iPLEX assay by the MassARRAY system. The first primer set was the barcode set. This set contained two primers: ITS5A and ITS4 (Table 6). This primer pair functioned as the amplification primer to specifically amplify the ITS region. The second set of primer is called the specific primer set. The set contained four primers (P1F, P1R, P2F, and P2R) (Table 6), which were designed by the Assay Design Suite (ADS), an online tool offered by Agena Bioscience (https://www.agenabio.com/services/assays-by-agena/) (accessed on 11 March 2024). The aim of this primer design was to cover each specific SNP site within the internal transcribed spacer 1 or ITS1 (primers: P1F and P1R) and the ITS2 (primers: P2F and P2R) regions. The primer pair between P1F and P1R provided the PCR product encompassing the ITS1 region (SNP 18 and 112). Similarly, the primers P2F and P2R targeted the ITS2 region and covered SNP sites: 577, 577, and 623 (Figure 4A). The iPLEX extension primer set was specifically designed for the iPLEX extension assay, the final step of the species identification (Table 6). The iPLEX extension primer set, P1#1, P1#2, P2#1, P2#2, and P2#3, which were designed specifically to each SNP positions 18, 112, 577, 623, and 652, respectively, were used in the single-base extension (SBE) reaction (Figure 4B, Table 6).

### 4.3. DNA Extraction, Amplification, and Nucleotide Sequencing

The leaves of authenticated *P. indica*, *P. zeylanica*, and *P. auriculata*, as well as commercial crude drug samples, were wiped with 75% ethanol to prevent fungal contamination. Genomic DNA was extracted from leaf specimens, crude drugs, and three different Thai traditional formulations using the DNeasy Plant Mini Kit (Qiagen, Hilden, Germany), following the manufacturer’s instructions with minor modifications [33]. The DNA concentration was measured using a NanoDrop 2000C Spectrophotometer (Thermo Scientific, Waltham, MA, USA), while the quality of the extracted DNA was assessed through agarose gel electrophoresis. Genomic DNA was electrophoresed in a 1.8% agarose gel and visualized under UV light using a Gel Doc EZ Imager (Bio-Rad, Hercules, CA, USA). Only high-quality genomic DNA with an A260/A230 ratio greater than 1.5 was selected for further analysis.

The ITS region of the authentic species was amplified from the extracted genomic DNA, and DNA sequencing was performed for DNA barcode analysis. Briefly, 100–120 ng of total DNA were used as the template in a 25 µL reaction mixture containing 1× PCR buffer for KOD FX Neo, 0.2 mM dNTPs, 0.2 µM of ITS5A and ITS4 primers, and 0.5 U of KOD FX Neo (TOYOBO Life Science, Osaka, Japan). PCR amplification was conducted using a T960 Thermal Cycler (Drawell, Chongqing, China) with the following cycling conditions: 94 °C for 2 min, followed by 35 cycles of 94 °C for 15 s, 53 °C for 30 s, and 68 °C for 45 s, with a final cycle at 68 °C for 5 min. The PCR product was electrophoresed on a 1.8% agarose gel, stained with GelRed (Biotium, Fremont, CA, USA), and visualized under UV light using a Gel Doc EZ Imager (Bio-Rad, Hercules, CA, USA). Successful PCR amplicons were purified using the MEGAquick-spin Plus Total Fragment DNA Purification Kit (Intron Biotechnology, Seongnam, Republic of Korea) and subsequently bidirectionally sequenced using an ABI PRISM 3730XL sequencer (Applied Biosystems, Waltham, MA, USA). BioEdit version 7.0.5 [50] was used to manually trim and edit the raw nucleotide sequences. The sequences were then aligned with published nucleotide sequences obtained from the NCBI database using the MUSCLE program in Molecular Evolutionary Genetics Analysis (MEGA 11) software, version 11.0.13 [51]. Single-nucleotide polymorphism (SNP) sites were identified and used for further analysis.

### 4.4. The iPLEX Assay on the MassARRAY System for the P. indica Identification

The workflow of the iPLEX assay on the MassARRAY system is shown in Figure 4A–C. Five SNP positions (18, 112, 577, 623, and 652) on the ITS region of *P. indica* were targeted for multiplex PCR and iPLEX single-base extension processes. The Mass spectra of the iPLEX extension products were predicted (Table 1). The PCR reaction was performed in 384-well plates using a 5 µL PCR cocktail, which contained 10 ng/µL total DNA as the template, 1× PCR buffer, 2 mM MgCl_2_, 500 µM dNTP mix, 100 nM each amplification forward and reverse primers, and 0.2 U of PCR enzyme (Agena Bioscience, San Diego, CA, USA). The PCR conditions were as follows: 95 °C for 2 min, followed by 45 cycles of 95 °C for 30 s, 56 °C for 30 s, and 72 °C for 60 s, with a final extension at 72 °C for 5 min. The reaction plates were sealed and stored at −20 °C for further analysis.

The iPLEX single-base extension reaction was conducted following the manufacturer’s instructions. Briefly, shrimp alkaline phosphatase (SAP) was added to the PCR product solution to eliminate excess dNTP residues. The SAP reaction mixture (2 µL) contained 0.17 µL of SAP Buffer (10×), 0.30 µL of SAP enzyme (1.7 U/µL), and 1.53 µL of H_2_O. The SAP solution mixture was combined with the PCR product. The reaction plates were sealed and incubated at 37 °C for 40 min, followed by 85 °C for 5 min. After incubation, the SAP-treated reaction mixtures were subjected to the primer extension reaction using the iPLEX Gold assay (Sequenom Inc., San Diego, CA, USA). The iPLEX extension reaction was performed in 2 µL, consisting of 0.62 µL of distilled water, 0.2 µL each of 10× iPLEX buffer plus and 10× iPLEX termination mix, 0.04 µL of iPLEX Pro enzyme, and 0.94 µL of the extension primer. The extension reaction was carried out under the following conditions: 95 °C for 30 s, followed by 40 cycles of 95 °C for 5 s, 52 °C for 5 s, and 80 °C for 5 s, with a final extension at 72 °C for 3 min. The extension products were dispensed onto the SpectroCHIP Array using an automated nanodispenser and subsequently analyzed by MALDI-TOF mass spectrometry on the MassARRAY platform (Figure 4C). The results were analyzed using SpectroTYPER version 4.0 software. Mass spectra of the samples were interpreted and compared with the predicted masses of the iPLEX extension products (Table 1).

### 4.5. Sensitivity of the iPLEX Extension Assay for the Differentiation of P. indica from Its Related Species

Sensitivity analysis was performed by preparing a series of 10-fold dilutions. Nine DNA concentrations were prepared: 10, 1, 0.1, 0.01, 0.001, 0.0001, 0.00001, 0.000001, and 0 ng/µL. The iPLEX extension reaction and mass spectra measurement using the MassARRAY system were carried out as described above. The limit of detection (LOD) was determined by analyzing the lowest DNA concentration at which allele-specific spectra could be accurately detected and interpreted as positive signals in the spectra.

### 4.6. Identification of P. indica in Plumbago-Mixed Samples

To evaluate the ability of the iPLEX extension reaction for detecting *P. indica* in *Plumbago*-mixed samples, genomic DNA from each *Plumbago* species was extracted, and two- and three-species mixed samples were prepared. The DNA mixtures were combined in a 1:1 (*v*/*v*) ratio. The samples, namely *P. indica* + *P. zeylanica* (PI + PZ), *P. indica* + *P. auriculata* (PI + PA), *P. zeylanica* + *P. auriculata* (PZ + PA), and *P. indica* + *P. zeylanica* + *P. auriculata* (PI + PZ + PA), were then analyzed using the iPLEX extension assay. The resulting mass spectra were interpreted using the MassARRAY system.

### 4.7. Authentication of the Commercial P. indica Crude Drugs and the Polyherbal Products

Ten commercial medicinal roots of *P. indica* or Jettamoon Pleung Daeng and three Thai traditional polyherbal products, namely Ya Benchakun, Ya Fai Ha Gong, and Ya Fai Pralaikan (Table 1), were tested using the iPLEX extension assay on the MassARRAY system to identify *Plumbago* species in the samples. Genomic DNA was extracted from the crude drugs and polyherbal formulations following the protocol outlined in the genomic DNA extraction section. The extracted DNA was then purified using the OneStep PCR Inhibitor Removal Kit (Zymo Research, Irvine, CA, USA) according to the manufacturer’s protocol. The iPLEX extension and mass spectra measurement steps were performed as described in the previous section.

## 5. Conclusions

In this study, we developed and validated a MassARRAY iPLEX assay for differentiating three *Plumbago* species—*P. indica*, *P. zeylanica*, and *P. auriculata*—by targeting nucleotide variations within the ITS DNA barcode region. Specifically, we focused on SNP sites located at positions 18, 112, 577, 623, and 652. The assay exhibited high sensitivity, enabling accurate identification of *P. indica* and distinguishing it from closely related species at DNA concentrations as low as 0.01 ng/µL. The assay was successfully applied to nine commercial *P. indica* root samples, identifying two as *P. zeylanica*. Furthermore, it demonstrated the ability to detect *P. auriculata* in mixtures with *P. indica*. However, the method showed limitations in distinguishing *P. zeylanica* from *P. indica* in mixed samples, likely due to the overlap in the mass spectra of these species at certain SNP positions. These findings confirm that the developed assay is a reliable and valuable tool for the authentication of *P. indica* raw materials.

## Figures and Tables

**Figure 1 ijms-26-07168-f001:**
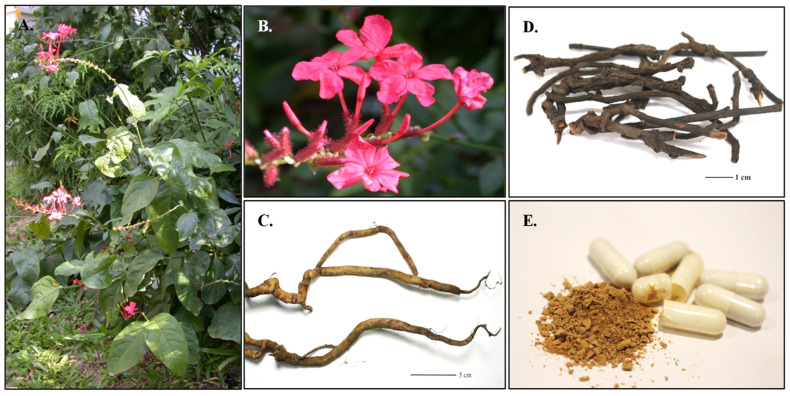
Morphological Characteristics and Utilization of *Plumbago indica* L.: (**A**). habit, (**B**). flowers, (**C**). roots, (**D**). Jettamoon Pleung Daeng crude drugs derived from dried *P. indica* roots, (**E**). Ya Benchakun formulation.

**Figure 2 ijms-26-07168-f002:**
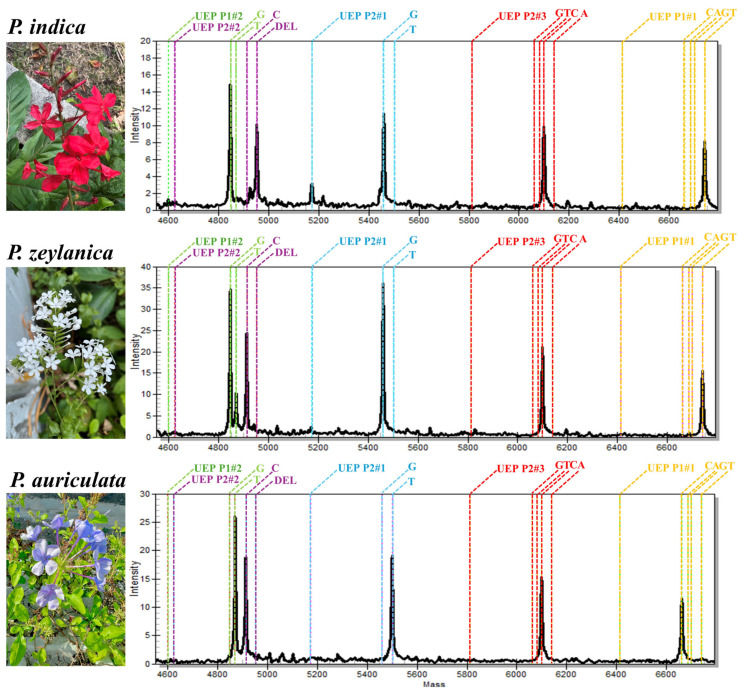
Mass spectra of authentic *Plumbago* species. The colors represent the mass profiles of the unextended primers (UEPs) and their corresponding base extension products: yellow for P1#1, green for P1#2, blue for P2#1, purple for P2#2, and red for P2#3.

**Figure 3 ijms-26-07168-f003:**
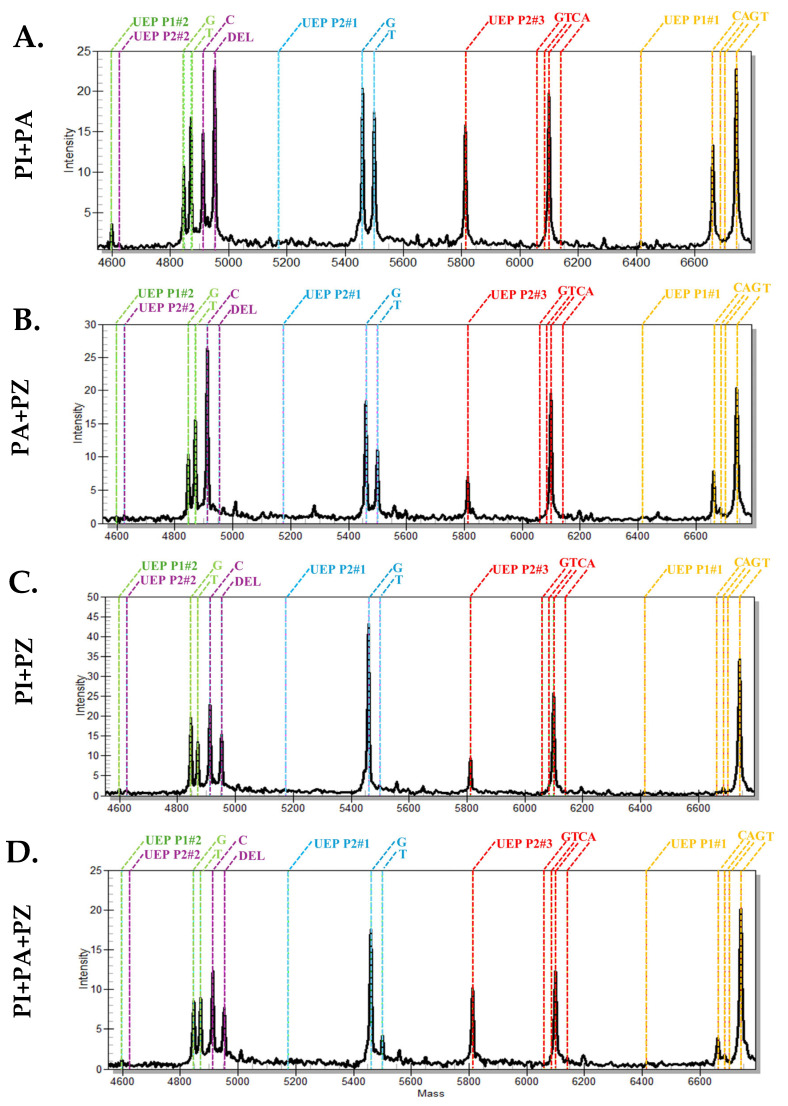
MALDI-TOF mass spectral fingerprint of *Plumbago*-mixed samples containing *P. indica* (PI), *P. zeylanica* (PZ), and *P. auriculata* (PA): (**A**) *P. indica* and *P. auriculata*, (**B**) *P. auriculata* and *P. zeylanica*, (**C**) *P. indica* and *P. zeylanica*, (**D**) *P. indica*, *P. auriculata*, and *P. zeylanica*. The colors represent the mass profiles of the unextended primers (UEPs) and their corresponding base extension products: yellow for P1#1, green for P1#2, blue for P2#1, purple for P2#2, and red for P2#3.

**Figure 4 ijms-26-07168-f004:**
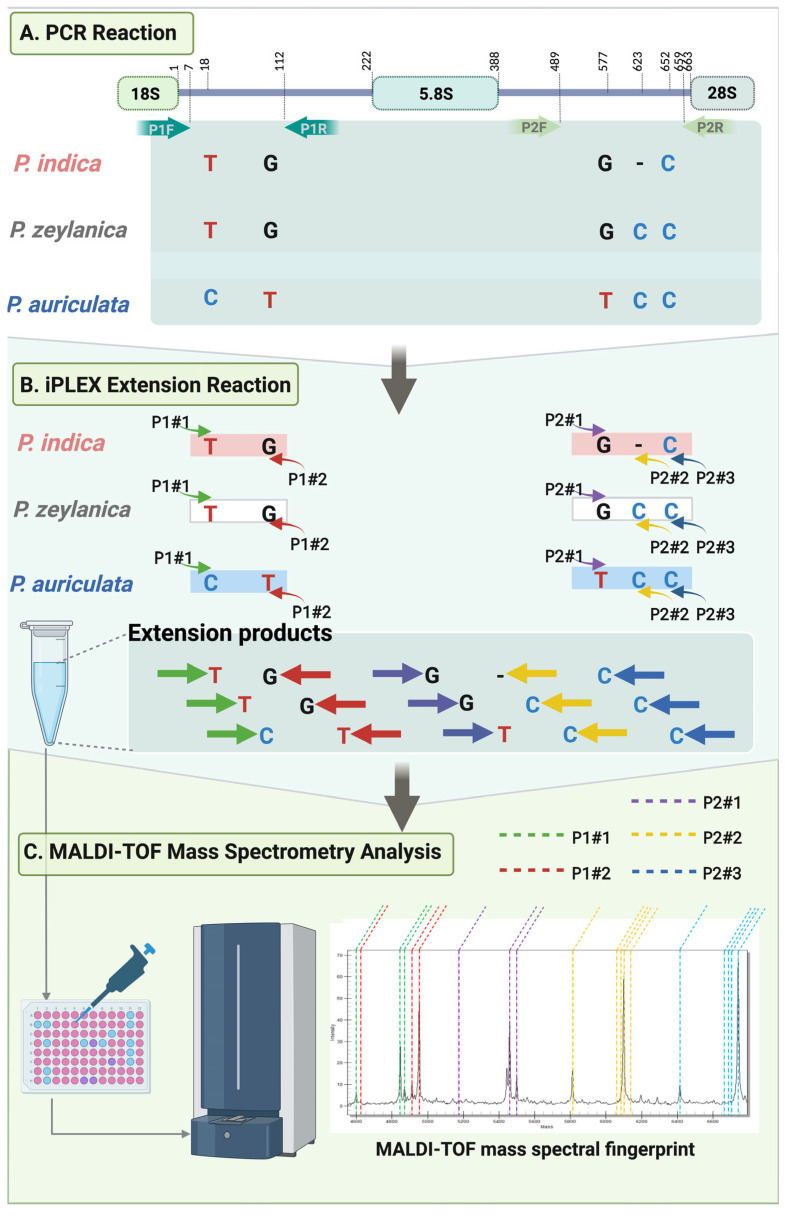
Molecular Principles and Workflow of the MassARRAY Method and Primer Design Concept for Differentiating Three *Plumbago* Species. The colors represent the arrows and mass profiles of the unextended primers (UEPs) and their corresponding base extension products: green for P1#1, red for P1#2, purple for P2#1, yellow for P2#2, and blue for P2#3. Created using BioRender (Intharuksa, A., 2025). Available online: https://BioRender.com/nu2pugl (accessed on 5 June 2025).

**Table 1 ijms-26-07168-t001:** Nucleotide sequences of the single-base extension primers and their base extension products used for the detection of SNPs in ITS region of *Plumbago* species.

Primers	Target SNP Position	Primer Sequences (UEP)	Mass (Da)	Expected Masses of the Single-Base Extension Product (Da)				
				C	A	G	T	DEL
P1#1	18	AAGGATCATTGTCGAAACCTC	6414.2	6661.4	6685.4	6701.4	6741.3	n.d.
P1#2	112	TTGTTCAAGCCTGGG	4599.0	n.d.	n.d.	4846.2	4870.2	n.d.
P2#1	577	CCGCGAAGCGTCGTGCC	5172.4	n.d.	n.d.	5459.6	5499.5	n.d.
P2#2	623	CCTGGGGTCGCATGG	4625.0	4912.2	n.d.	n.d.	n.d.	4952.1
P2#3	652	ATATGCTTAAACTCAGCGG	5811.8	6059.0	6083.0	6741.3	6138.9	n.d.

UEP = unextended primer; Da = Dalton; DEL = deletion; n.d. = not detected.

**Table 2 ijms-26-07168-t002:** The expected mass and the obtained mass profiles from the single-base extension product of *Plumbago* species.

Species	Target SNP Position	Expected Mass of the Single-Base Extension Products				Mass Profile of the Single-Base Extension Products from Authentic Plants			
		C	G	T	INDEL	C	G	T	INDEL
*P. indica*	P1#1	-	-	6741.3	-	-	-	6741.3	-
	P1#2	-	4846.2	-	-	-	4846.2	-	-
	P2#1	-	5459.6	-	-	-	5459.6	-	-
	P2#2	-	-	-	4952.1	-	-	-	4952.1
	P2#3	6059.0	-	-	-	6059.0	-	-	-
*P. zeylanica*	P1#1	-	-	6741.3	-	-	-	6741.3	-
	P1#2	-	4846.2	-	-	-	4846.2	4870.2 *	-
	P2#1	-	5459.6	-	-	-	5459.6	-	-
	P2#2	4912.2	-	-	-	4912.2	-	-	-
	P2#3	6059.0	-	-	-	6059.0	-	-	-
*P. auriculata*	P1#1	6661.4	-	-	-	6661.4	-	-	-
	P1#2	-	-	4870.2	-	-	-	4870.2	-
	P2#1	-	-	5499.5	-	-	-	5499.5	-
	P2#2	4912.2	-	-	-	4912.2	-	-	-
	P2#3	6059.0	-	-	-	6059.0	-	-	-

* Minor peak. INDEL = an insertion/deletion.

**Table 3 ijms-26-07168-t003:** Mass spectral results for the sensitivity analysis of the method used in this study.

Samples	DNA Concentration (ng/μL)	Extension Primers				
		P1#1	P1#2	P2#1	P2#2	P2#3
*P. indica*	10	T	G	G	DEL	C
	1	T	G	G	DEL	C
	0.1	T	G	G	DEL	C
	0.01	T	G	G	DEL	C
	0.001	T	G	G	C/DEL	C
	0.0001	T	G/T	G	C	C
	0.00001	T	G	G	C	C
	0.000001	T	G/T	G	C	C
	0	-	-	-	-	-
*P. zeylanica*	10	T	G	G	C	C
	1	T	G	G	C	C
	0.1	T	G	G	C	C
	0.01	T	G	G	C	C
	0.001	T	T	G	C	C
	0.0001	T	G/T	G	C	C
	0.00001	T	G	G	DEL	C
	0.000001	T	G/T	G	C	C
	0	-	-	-	-	-
*P. auriculata*	10	C	T	T	C	C
	1	C	T	T	C	C
	0.1	C	T	T	C	C
	0.01	C	T	T	C	C
	0.001	C/T	G/T	G/T	C	C
	0.0001	T	G	G	C	C
	0.00001	T	G/T	G	C	C
	0.000001	T	G/T	G	C	C
	0	-	-	-	-	-

DEL = deletion.

**Table 4 ijms-26-07168-t004:** Mass spectral results for species identification of authentic *Plumbago* species, commercial Jettamoon Pleung Daeng crude drugs, and polyherbal formulations.

Samples	Extension Primers					Result
	P1#1	P1#2	P2#1	P2#2	P2#3	
PI	T	G	G	DEL	C	*P. indica*
PZ	T	T	G	C	C	*P. zeylanica*
PA	C	T	T	C	C	*P. auriculata*
C-1	T	G	G	DEL	C	*P. indica*
C-2	T	G	G	DEL	C	*P. indica*
C-3	T	G	G	DEL	C	*P. indica*
C-4	T	G	G	DEL	C	*P. indica*
C-5	T	G	G	DEL	C	*P. indica*
C-6	T	G	G	DEL	C	*P. indica*
C-7	T	G	G	DEL	C	*P. indica*
C-8	T	G	G	DEL	C	*P. indica*
C-9	T	G	G	DEL	C	*P. indica*
R-1	T	G	G	DEL	C	*P. indica*
R-2	T	T	G	C	C	*P. zeylanica*
R-3	T	G	G	DEL	C	*P. indica*
R-4	T	G	G	DEL	C	*P. indica*
R-5	T	G	G	DEL	C	*P. indica*
R-6	T	T	G	C	C	*P. zeylanica*

DEL = deletion.

**Table 5 ijms-26-07168-t005:** Plant materials, crude drugs, and Thai traditional formulation used in this study.

Codes	Sample Details	Locality
Authentic *Plumbago* species		
PI	*P. indica* L.	Maerim, Chiang Mai
PZ	*P. zeylanica* L.	Mueang, Chiang Mai
PA	*P. auriculata* Lam.	Mueang, Chiang Mai
Crude drugs		
C-1	Jettamoon Pleung Daeng	Mueang, Phatthalung
C-2	Jettamoon Pleung Daeng	Samphanthawong, Bangkok
C-3	Jettamoon Pleung Daeng	Samphanthawong, Bangkok
C-4	Jettamoon Pleung Daeng	Samphanthawong, Bangkok
C-5	Jettamoon Pleung Daeng	Mueang, Nakhon Pathom
C-6	Jettamoon Pleung Daeng	Hat Yai, Songkhla
C-7	Jettamoon Pleung Daeng	Mueang, Nakhon Pathom
C-8	Jettamoon Pleung Daeng	Samphanthawong, Bangkok
C-9	Jettamoon Pleung Daeng	Mueang, Chiang Mai
Thai traditional formulations		
R-1	Ya Benchakun	Faculty of Pharmacy, Chiang Mai University (In-house preparation)
R-2	Ya Benchakun	Company 1
R-3	Ya Fai Pralaikan	Faculty of Pharmacy, Chiang Mai University (In-house preparation)
R-4	Ya Fai Pralaikan	Company 2
R-5	Ya Fai Ha Kong	Faculty of Pharmacy, Chiang Mai University (In-house preparation)
R-6	Ya Fai Ha Kong	Company 3

**Table 6 ijms-26-07168-t006:** Nucleotide sequences of the primers used in this study.

Primer Set	Primer Name	Target(Region/SNP)	Direction	Primer Sequences (5′→ 3′)	Original Design
DNA barcode	ITS5A	ITS region	Forward	CCT TAT CAT TTA GAG GAA GGA G	[48]
	ITS4	ITS region	Reverse	TCC TCC GCT TAT TGA TAT GC	[49]
Specific primer	P1F	ITS1 region	Forward	ACG TTG GAT GAA CCT GCG GAA GGA TCA TTG	This study
	P1R	ITS1 region	Reverse	ACG TTG GAT GGC GCC GTG TTT TTG TTC AAG	This study
	P2F	ITS2 region	Forward	ACG TTG GAT GCG GTT GGC TTA AAT TCG GG	This study
	P2R	ITS2 region	Reverse	ACG TTG GAT GCT TAT TGA TAT GCT TAA ACT	This study
iPLEX extension primer	Ext_P1#1	SNP18	Forward	AAG GAT CAT TGT CGA AAC CTC	This study
	Ext_P1#2	SNP112	Reverse	TTG TTC AAG CCT GGG	This study
	Ext_P2#1	SNP577	Forward	CCG CGA AGC GTC GTG CC	This study
	Ext_P2#2	SNP623	Reverse	CCT GGG GTC GCA TGG	This study
	Ext_P2#3	SNP652	Reverse	ATA TGC TTA AAC TCA GCG G	This study

## Data Availability

Data will be made available on request.

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
