# Peer review of "Unveiling Adulteration in Herbal Markets: MassARRAY iPLEX Assay for Accurate Identification of Plumbago indica L."

_ijms, 2025, doi:10.3390/ijms26157168_

Round 1
Reviewer 1 Report
Comments and Suggestions for Authors
The manuscript dealing with identifying adulteration in Herbal markets using a MassArray IPLEX Assay is quite interesting and the idea is worth publishing. The manuscript itself and the study however could benefit from a few improvements to raise the scientific interest and get it maybe a little bit above the pure method presentation. Let me explain a bit further::
1.) If someone wants to belittle the manuscript it could be argued that the only improvement of the method is to replace the sequencing of the PCR products by MALDI-TOF MS. For comparison it would be beneficial to see the sensitivity of both methods on the same products i would therefore recommend to create a dilution of the PCR products and send the same fragments to traditional sequencing vs. the iplex method to compare comparison. Afterwards you could give a statement of the sensitivity, the pure time consuming step is not convincing enough since those also highly depend on the skills of the user handling the blast search etc.
2.) The therapeutic relevance of the method as described in the method is at least a little bit overstated. As far as i could get the method only detects the presence of non-desired Plumbagio species. But specially for therapeutic uses the most important factor is the quantity of the adulterated herbal material. I could therefore imagine a screening of the bioactive compounds and quantitation could deliver an insight if the sensitivity of the method is too high or too low to give therapeutic relevant insights into the quality of the Plumbago.
3.) In the discussion part the manuscript stays often by rephrasing results instead of discussing them to the wider meaning for the scientific community. Please rework this discussion part.
In total i would recommend a major revision of the manuscript to improve the quality and the impact of the work presented.
Reviewer 2 Report
Comments and Suggestions for Authors
This manuscript addresses a critical issue in herbal medicine quality control: the adulteration of medicinal plant raw materials. The authors developed and applied a MassARRAY iPLEX SNP genotyping assay to identify Plumbago indica and differentiate it from closely related species. The approach is technically sound and presents practical applications. However, several important scientific and methodological limitations need to be addressed before the work can be considered for publication.
1. Section 2.4, pp. 5–6. The authors rely solely on MassARRAY data to confirm species identity in crude drugs and formulations. No orthogonal techniques, such as HPTLC, HPLC, or conventional DNA sequencing, were applied to independently verify adulteration cases. At least one independent method (e.g., chemical profiling or Sanger sequencing) should be used to validate the results.
2. Section 2.3, p. 4. While species substitution is qualitatively detected, the assay does not attempt to quantify the relative proportion of adulterants in mixed samples. Consider estimating relative abundances based on peak intensities or standard curves.
3. Table 1, Figure 3. The rationale for selecting the five ITS SNP positions (18, 112, 577, 623, and 652) is not clearly explained. Notably, SNP112 showed cross-reactivity between P. auriculata and P. zeylanica. Provide prior sequencing data or alignment analysis to justify SNP selection and assess inter-species variation.
4. Table 4, Figure 3. Although the assay distinguishes species in mixed samples, some SNP peaks are shared among species (e.g., SNP112 detected in both P. zeylanica and P. auriculata). The implications of such overlaps are not thoroughly discussed. A specificity/sensitivity analysis of each SNP marker and a discussion of possible cross-reactivity are necessary.
5. Section 4.7. There is no discussion of DNA degradation in industrial products, which is a known limitation in DNA-based herbal authentication. Include a test case or a discussion on the method’s performance with degraded or thermally processed samples.
6. No statistical metrics (e.g., standard deviation, confidence intervals, or repeatability coefficients) are provided for the SNP detection results. Include statistical validation of the assay performance.
7. Terms such as “INDEL” are used in tables and figures without prior definition, which may confuse non-specialist readers.
8. Adding a short paragraph in the introduction based on recent studies on identifying adulteration in herbal medicinal products can improve the quality of the article. Here, references to some articles, such as the following, can be useful.
1- Tarassoli, Z., et al. Hollow Fiber-Based Liquid Phase Microextraction .... J Anal Chem 79, 1717–1723 (2024). https://doi.org/10.1134/S1061934824701284
2- Dubnicka M, et al.. Investigation of the Adulteration of... 2020; 16(8): 965-969. [DOI:10.2174/1573411015666191127093710.
3- Shahbazi, H. et al., Evaluation of adulteration in Lavandula angustifolia Mill...J. Med. Plants 2021, 20(80): 34-46. 10.52547/jmp.20.80.34
Round 2
Reviewer 2 Report
Comments and Suggestions for Authors
The quality of the manuscript improved and became acceptable after the revision.